# A Simple Pattern of Movement Is Not Able to Inhibit Experimental Pain in FM Patients and Controls: An sLORETA Study

**DOI:** 10.3390/brainsci10030190

**Published:** 2020-03-24

**Authors:** Eleonora Gentile, Katia Ricci, Eleonora Vecchio, Giuseppe Libro, Marianna Delussi, Antonio Casas-Barragàn, Marina de Tommaso

**Affiliations:** 1Applied Neurophysiology and Pain Unit, SMBNOS Department, Bari Aldo Moro University, Polyclinic General Hospital, 70121 Bari, Italy; katiari86@gmail.com (K.R.); eleonora.vecchio@gmail.com (E.V.); giuseppelibro@libero.it (G.L.); marianna.delussi@uniba.it (M.D.); marina.detommaso@uniba.it (M.d.T.); 2Physical Therapy Department, University of Granada, 18016 Granada, Spain; antoniocb@ugr.es

**Keywords:** laser-evoked potentials, chronic pain, motor activity

## Abstract

Motor cortex activation seems to induce an analgesic effect on pain that would be different between patients with fibromyalgia (FM) and control subjects. This study was conducted to analyze the changes of the laser-evoked potentials (LEPs) induced during a finger tapping task in the FM patients and the controls employing a multi-dipolar analysis according to Standardized low resolution brain electromagnetic tomography (sLORETA) method. The LEPs from 38 FM patients and 21 controls were analyzed. The LEPs were recorded while subjects performed a slow and a fast finger tapping task. We confirmed that the difference between N1, N2 and P2 wave amplitudes between conditions and groups was not significant. In control subjects, the fast finger tapping task induced a modification of cortical source activation in the main areas processing laser stimulation from the moving hand independently from the movement speed. In summary, a simple and repetitive movement is not able to induce consistent inhibition of experimental pain evoked by the moving and the not moving hand in each group. It could interfere with LEP sources within the limbic area at least in control subjects, without inhibit cortical responses or explain the different pattern of motor and pain interaction in FM patients.

## 1. Introduction

Pain and motor activation are strictly interrelated, as the expression of one could modify the other for a reciprocal influence. This is partly confirmed by the reducing effect on the chronic pain exerted by the stimulation of the primary motor cortex [1] and by the exercise-induced analgesia [2]. Laser-evoked potentials are a useful tool in the study of pain pathways and they are modulated under different conditions, including concurrent movement [3]. The vertex potential is related to the execution of defensive actions [4], so it could be reduced during a concurrent motor engagement. Fibromyalgia is a chronic condition with diffuse pain, sleep disturbances, fatigue, and several associated symptoms, as detailed in the most recent criteria [5]. Motor activity is deficient in such patients [6,7], while correct movement and exercise could contribute to a better condition [8,9]. Moreover, the activation of motor cortex, which is induced by noninvasive neurostimulation, might reduce symptoms severity [10,11]. In a recent study, we evaluated motor cortex metabolism and laser-evoked potentials during a finger tapping task in patients with fibromyalgia (FM) and in control subjects [12]. We found a slower finger tapping speed and reduced activation of motor cortex in FM patients, while LEPs obtained with the moving and the not moving hand appeared scarcely modified during the execution of the finger tapping task in both patients and controls [12]. We thus concluded that patients with FM have reduced efficiency in movement and motor cortex activation, as confirmed by our previous studies. The lack of a modulatory effect of the finger tapping task on the amplitude of laser cortical responses, which was also evident in control subjects, appeared less justified and in apparent contradiction with previous studies [3]. A simple and repetitive movement such as a finger tapping task could have low influence on the cortical circuits subtending LEPs, so our hypothesis was that only movements employing the motor program and cognitive engagement could interfere with the salience of the painful stimulus [13]. This could be relevant in the design of the motor exercises capable of improving the condition. We aimed to further analyze the LEPs recorded in the FM patients and in the control subjects in the basal conditions and during the finger tapping task, reconsidering the data obtained in our recent study [12] with topographical and statistical examination, and conduct a source analysis using the LORETA-KEY V20190617 software. Our hypothesis is that subtle changes could occur in topographical distribution and cortical area activation, explaining differences in motor and pain processing interaction between patients and control subjects.

## 2. Materials and Methods

### 2.1. Subjects

The experimental procedure and case selection are reported in our recent study [12]. The participants were thirty-eight FM patients (35 females, 3 males, mean age 42.8 ± 10.163 years, mean ± SD) and twenty-one control subjects (13 females, 8 males, mean age 32.62 ± 13.912 years, mean ± SD) recruited from the Applied Neurophysiology and the Pain Unit of the Bari University. The average duration of the disorder for FM patients was 5.48 years. The severity level of pain measured by the widespread pain index (WPI) was M = 12.40 and SD = 4.85 (maximum score: 19, cutoff: 7) [14]. The patient group was homogeneous in terms of comorbidities and pain symptoms in the basal condition. All the patients were submitted to a detailed interview reporting the issues considered valid for FM criteria according to the 2010/2011 American College of Rheumatology criteria [5] and according to the International Association for the Study of Pain (IASP) classification of chronic pain for International Classification of Diseases (11^th^ revision) (ICD-11) [15], including diffuse pain and associated symptoms, such as fatigue, sleep disturbances, and cognitive impairment. All the cases presenting with the conditions reported in the exclusion criteria were not admitted to the present study. The Ethics Committee of the Bari Polyclinic’s General Hospital approved the experimental study (Ethical Approval Code: 5902). All right-handed participants were included in the study in accordance with the Edinburgh Handedness Inventory [16]. Written informed consent was given by all participants prior to their taking part in the study. The exclusion criteria defined for participation in the experimental study were the presence of central and peripheral nervous system disorders, rheumatic diseases, current or previous history of cancer, psychiatric symptoms, and intake of substances affecting the central nervous system. All the FM patients underwent a clinical evaluation before being recruited to the experimental study. The patients started the drug treatment prescribed by the clinician after participation in the experiment.

### 2.2. Experimental Procedure

The participants were asked to sit in a comfortable chair and remain relaxed throughout the experiment. The detailed experimental protocol is reported by Gentile et al. [12] and shown in Figure 1. The order of experimental conditions was randomized. The experimenter trained the participants to perform a finger tapping task that consisted of clicking a pushbutton panel slowly or as fast as possible. The motor task was executed with the thumb of the right hand. First, the participants were asked to stay relaxed, carefree, and stare at a cross in the middle of a black screen on the computer monitor to record the two minutes of the resting state. The experimenter used a laser stimulator to elicit a nociceptive response in the participants. The laser stimulus was delivered on the right- or left-hand dorsum for each participant. Noxious laser stimuli were applied at a fixed interval of 10 s. For each experimental condition, we delivered 30 laser stimuli on the left hand or on the right hand of a participant. Participants were asked to count the perceived laser stimuli to maintain a high level of attention. Therefore, participants had to stay still and to focus on the finger tapping task. As far as the slow finger tapping task (SFT) is concerned, the participants had to click a button every five seconds, as indicated by the experimenter. To perform the fast finger tapping (FFT) task, they were required to click the button as quickly as possible. Each finger tapping task was also performed during the noxious laser stimulation of both the right hand and the left hand. In addition, we decided to stimulate the non-moving left hand to explore the possible distractor effect of movement on pain perception. As a result, to evaluate the motor performance of participants, the speed of the finger tapping task was computed by counting the number of clicks per second. The interval between each experimental condition was fixed at 60 s.

### 2.3. EEG Recording

The EEG was recorded using a cap with 61 scalp electrodes positioned according to the 10–20 International system. The reference electrode was positioned at the nasion, and the ground electode—at the Fpz. We used a Micromed System Plus (Mogliano Veneto, Italy) to amplify the EEG signal with a sampling frequency of 256 Hz. The electrooculogram was recorded using two electrodes placed over the lower right and left eyelids. Electrode impedances were kept below 5 kΩ. The signal was digitalized using a 0.1–70 Hz range filter, and the notch filer was fixed at 50 Hz.

Nociceptive laser stimulation. We delivered nociceptive stimuli using CO_2_ laser pulses (wavelength, 10.6 mm; beam diameter, 2 mm, Neurolas Electronic Engineering, Florence, Italy). The inter-stimulus interval was fixed at 10 s. For each participant, we evaluated the subjective pain threshold according to the method of the limits, and we delivered laser stimuli when the subjects felt a pinprick sensation [17]. The experimenter asked each participant to rate his/her pain on a numerical rating scale (NRS) [18]. After each laser stimulation, the subjects were required to rate the perceived intensity of their pain on the visual analogue scale (VAS) [19]. The “0” value on the scale indicated “no pain,” and the maximal value “100” represented “unbearable pain.”

### 2.4. EEG Analysis

Preprocessing was performed in MATLAB using the EEGLAB 14_1_1 tool. The data were first high-pass filtered at 1 Hz to remove slow drifts. Next, a notch filter at 50 Hz (L: 48, H: 52) was applied to remove power line noise artifacts. Artifact components were then automatically removed considering the components recorded on the electrooculogram (EOG) channels. Bad channels were identified by a semiautomatic method based on visual detection and channel statistics. To precompute channel measures, spherical interpolation of missing channels and deletion of Independent Component Analysis (ICA) artifact components pre-tagged in each dataset was performed. Channels presenting with distributions of potential values further away from the Gaussian distribution than other scalp channels were also removed. In both groups, we deleted 1 EEG segment for each recording session on the average. Laser-evoked potentials (LEPs) were precomputed in the time interval ~100–1000 ms using a 70 Hz low-pass filter, removing the baseline and considering the 100 ms preceding the laser stimulus. After a visual analysis of LEPs, we precomputed the event-related response amplitude in the total considered time interval, and statistical probability maps (SPMs) were obtained in the 700 ms (420 time frames at 256 Hz) following the laser stimuli. In particular, we checked time intervals 150–180 ms for the N1, 200–250 ms for the N2, and 300–400 ms for the P2 component, taking into consideration the previous normative data [20] confirmed by visual analysis of single tracks.

### 2.5. Standardized Low-Resolution Brain Electromagnetic Tomography

The standardized low-resolution brain electromagnetic tomography (sLORETA) was used to generate the topographical analysis of LEPs (SAKA 2011 version) [13,14,15]. Previous experimental studies have supported the usefulness and validity of the standardized low-resolution brain electromagnetic tomography (sLORETA) in localizing generators of scalp-recorded potentials, including those related to pain processing and modulation [20,21,22,23,24]. LEPs in the ~100–700 ms intervals were only submitted to the sLORETA analysis. A randomization procedure for the statistical non-parametric maps (SnPM) was applied with 5000 randomizations according to the LORETA software. The randomization procedure was implemented to control for type I errors arising from multiple comparisons.

### 2.6. Statistical Analysis

Statistical analysis was performed applying the parametric statistic model of the EEGlab tool corrected for the Bonferroni multiple comparison test, with a 3 × 2 analysis of variance (ANOVA) model, where experimental conditions (laser stimulation condition on the right or on the left hand dorsum (basal), SFT + Laser stimulation of the right or left hand dorsum, FFT + Laser stimulation of the right or left hand dorsum) and two groups (patients vs. controls) were compared.

For the sLORETA analysis, statistical differences between the conditions were computed as images of voxel-by-voxel *t*-values. The localization of the differences in cortical activity was based on the standardized electric current density and resulted in three-dimensional *t*-score images. In these images, cortical voxels of statistically significant differences were identified by a non-parametric approach, with a 5% probability level threshold determined by 5000 randomizations [25]. For the sLORETA analysis, a *t*-test for paired groups was performed separately in the patient and control groups, comparing the basal vs. the finger tapping task conditions. Independent group analysis of sLORETA changes between the basal vs. the fast finger tapping task conditions was then run. We evaluated the correlation of the sLORETA matrix during the fast finger tapping task with the finger tapping speed using the regression analysis included in the sLORETA software.

## 3. Results

Finger tapping speed was slower in FM patients, as reported by Gentile et al. [12]. Therefore, motor performance was higher in the controls than in the patients regardless of the pain induced by laser stimuli (Table 1).

### 3.1. Right Hand

Laser-evoked response amplitudes. The P2 component seemed smaller in the patients compared to the controls. The same wave seemed reduced during the fast finger tapping task in healthy subjects, while this effect was less evident in the patients (Figure 2).

Even though the topographical maps also showed that the earlier N1 and N2 components were also reduced in amplitude during a slow and especially a fast movement, no statistical significance emerged (Figure 3, Figure 4 and Figure 5).

**Pain perception intensity.** The values on the VAS were similar for laser conditions and during finger tapping tasks. However, there was a significant difference in the intensity of pain perception between the group of patients and the group of controls. For details, see Table 1.

**sLORETA.** In the control group, the comparison of LEP amplitudes showed a slight decrease of voltage in time interval 300–350 ms (*t*-threshold for a large effect size: 0.25–3.48, *t*-values in the time interval of 300–350 ms ranging from 3.23–3.34 corresponding to an effect size of 0.15 and the medium Cohen’s *d*-value of 0.5). In the same group, the comparison between the basal condition and the fast movement (FFT) caused a significant reduction of cortical activation that reached statistical significance in the interval of 300–350 ms (*t*-threshold for large effect size: 3.48; Cohen’s d–value: 0.8). The reduction of sLORETA voxels was visible in several cortical regions (frontal, limbic, and insular regions—see Figure 6 and Appendix A).

In the patients, LEP amplitudes and sLORETA values were similar between the two considered conditions (basal vs. FFT + laser stimulation of the right hand dorsum).

The comparison between the groups showed only a medium-sized modification of cortical source activation in Broadman areas corresponding to temporal and limbic regions in the interval of 300–350 ms (*t*-threshold for a medium effect size: 3.10; effect size: 0.5; Cohen’s *d*-value: 0.5) (Figure 7, Appendix A).

### 3.2. Left Hand

**Laser-evoked potential amplitude**. The changes in amplitude of vertex LEPs for the stimulation of the left hand dorsum were not different between experimental conditions (Figure 8). In each group, there was a slight and not statistically significant decrease in the N1, the N2 and the P2 amplitude during the SFT and FFT tasks (Figure 9, Figure 10 and Figure 11). The statistical analysis performed on 61 channels did not show relevant differences (Figure 9, Figure 10 and Figure 11).

**sLORETA**. No relevant difference emerged from the statistical comparison of sLORETA values relative to the laser-evoked potentials obtained from the left hand stimulation in the basal condition and during a slow and a fast movement.

### 3.3. Correlations

No statistically relevant correlation emerged in the controls and the FM patients between the changes of the sLORETA values induced by fast movement and finger tapping speed.

## 4. Discussion

In this study, we substantially confirmed the negative results obtained in the previous analysis of LEP amplitude changes induced by a finger tapping task in FM patients and controls [12]. Slight modulation of the later vertex wave obtained by the stimulation of the moving hand was visible in the controls and caused a different activation of cortical sources located in the frontal and limbic regions. This was independent from the finger tapping speed. In the following paragraphs, we comment on the main points of this analysis.

*Absence of relevant LEP changes during a movement in the FM patients and the controls.* According to the previous analysis [12], we did not find significant modification of LEP amplitudes obtained from the moving and the not moving hand in the patients and the controls during the slow and the fast finger tapping tasks. The lack of effect on LEPs from the not moving hand could also suggest that this task did not exert a distraction sufficient to inhibit vertex waves. The LEPs are sensitive to the bottom-down influence exerted by relevant stimuli with a potentially dangerous effect on the body [26]. So far, a pure distractive effect induced by a repetitive movement of one’s own hand is largely improbable, as this movement is not relevant in view of potential danger and consequent motor reaction. The topographical maps of the laser-evoked components showed a modulation of the three main waves resulting from the stimulation of the moving and the not moving hand in the sense of their amplitude reduction and different scalp distribution during the slow and the fast finger tapping tasks. However, these changes were far from statistically relevant in the control and the FM groups.

The finger tapping task caused a significant change of motor cortex activation, more evident in the controls, as shown in our recent study [12]. There is evidence that stimulation of the primary motor cortex reduces pain symptoms in chronic patients [27]. It also seemed effective in reducing vertex LEPs in the control subjects and in migraine patients [28]. Moreover, finger tapping is a simple and repetitive movement provoking cerebellum activation [29]. The activation of the cerebellum induced by anodal transcranial direct current stimulation (TDCS) could inhibit LEP waves [30]. Considering these studies, we can suppose that the motor cortex and cerebellar activation induced by a simple repetitive movement did not resemble the effect induced by neurostimulation and is unable to inhibit LEPs. This lack of inhibition was similar in the controls and the FM patients, even though healthy subjects showed a consistent increase of cortical metabolism during the execution of the fast finger tapping [12].

In the studies exploring the effects of movement on the cortical responses evoked during concurrent nociceptive stimulation, it was the movement preparation and readiness to interfere with LEP generation [3,22]. Here, we confirm our hypothesis that execution of a simple repetitive movement is substantially unable to interfere with LEP waves, while cognitive engagement in a motor program and preparation could be necessary to contrast the substrate of LEP generation that is the cortical preparation to a defensive action against a salient stimulus [4].

A repetitive movement causes proprioceptive inference with the involvement of the primary and the secondary somatosensory cortex that would inhibit the cortical processing caused by the a delta fiber input [31,32]. The present results suggest that the somatosensory input generated by slow and fast repetitive movements did not inhibit LEPs in a relevant way.

Instead, a different scalp distribution emerged at least in the controls, as explained by the sLORETA analysis (see the paragraph below). Focal mechanical vibration was ineffective in reducing the LEPs obtained from the stimulated and the not stimulated hand in healthy volunteers [33]. Furthermore, the somatosensory inputs generated by a repetitive movement seem thus ineffective in reducing the amplitude of N1 devoted to the discrimination of pain features, and of N2 and P2 devoted to stimuli salience and motor reaction.

Subjective pain perception was also similar between the movement and the basal conditions, though it was different between the groups, as discussed in our previous study [12].

*The standardized low-resolution brain electromagnetic tomography* (*sLORETA) analysis*. The employment of the sLORETA software confirmed the lack of relevant modification of LEP amplitudes across conditions and between groups. Moreover, the analysis of a single comparison between basal conditions and the fast finger tapping showed some changes in cortical source activation at least in the controls and a tendency to a different pattern of cortical activation in the FM patients. The execution of the finger tapping task with the maximal possible speed seemed to reduce the activation of cortical regions within the limbic network in the anterior and posterior cingulate cortex, the precentral frontal lobe, and the insula. The time interval in which significant changes were present corresponded to the late vertex component, and the detected sources were compatible with the main generators of this wave, also in accord with the previous application of the sLORETA analysis to LEPs [24,34].

The contradiction between the amplitude analysis performed by an EEGlab MATLAB tool and the sLORETA analysis is only virtual. In fact, in topographical maps, the reduction of the P2 representation was evident over the frontal and the temporal regions in the controls in the condition of a fast finger tapping task, though it did not reach statistical significance in the comparison of the three conditions. This was the reason why we decided to extrapolate this single comparison aiming to emphasize this not striking phenomenon. Considering this analysis, we could only suggest that fast finger tapping, far from inducing clear inhibition of late LEPs, could thus modulate its cortical generators. We could thus suppose that repetitive movements, though devoid of relevant cognitive engagement, also have some effects on the late component of LEPs, just slightly reducing the strength of cortical activation in the regions devoted to the attribution of stimulus salience and consequent behavior [13]. This modulation seemed independent from movement efficiency, as no correlation emerged with finger tapping speed. In the FM patients, we did not observe any relevant LEP generator variation during fast movements, but the comparison of the sLORETA changes between the groups showed only a mild reduction of limbic and temporal regions activation in the P2 time interval. This result seems insufficient to claim for a different pattern of cortical activation during the fast finger tapping task in the FM patients in respect to the controls, but this trend seems in line with the reduced strength of motor activation observed in the FM patients [12] with a possible reduced efficiency in the modulation of nociceptive evoked responses.

The limited number of cases could have reduced the difference in cortical activation during concurrent finger tapping tasks between the FM patients and the control subjects. Moreover, further studies employing more complex motor tasks could clarify which modalities could better interfere with cortical pain processing.

## 5. Conclusions

The present results satisfied our primary hypothesis resulting from the observation of LEP topography during a fast finger tapping task in the cohort of healthy subjects recruited in the previously published study [12], as a reduction of cortical generators of the P2 component seemed to occur in this condition. It was not a purely distractive effect, because it was absent in the LEPs induced by the not moving hand, but it could be interpreted as an interference due to a stereotyped and repetitive movement to the cortical regions devoted to the decoding of stimuli salience. This phenomenon also seems independent from movement efficiency, as it was not related to the finger tapping speed, but was influenced by movement property, as it was absent during the slow task. We can also confirm our previously reported results about the lack of clear inhibition of LEPs due to this repetitive and stereotyped movement, because the effect we observed was limited as compared to the studies where more complex movements were tested [12].

No definite conclusion could be assumed regarding a possible different influence of finger tapping on the LEPs evoked in FM patients. LEP pattern is complex in FM patients, as it is influenced by different variables, peripheral nerve involvement, and central demodulation with reduced habituation [35]. The reduced efficiency in the finger tapping task and the defect in motor cortex activation [12] did not cause a clear difference in LEP modulation as compared to the control subjects. This could also confirm that a repetitive movement, such as finger tapping, could not emphasize the possible differences in interaction between motor performance and pain processing. Considering the potential importance of motor engagement in the management of FM patients [8,9], the present results outline the limited efficiency of a simple movement in modulating cortical pain processing.

## Figures and Tables

**Figure 1 brainsci-10-00190-f001:**
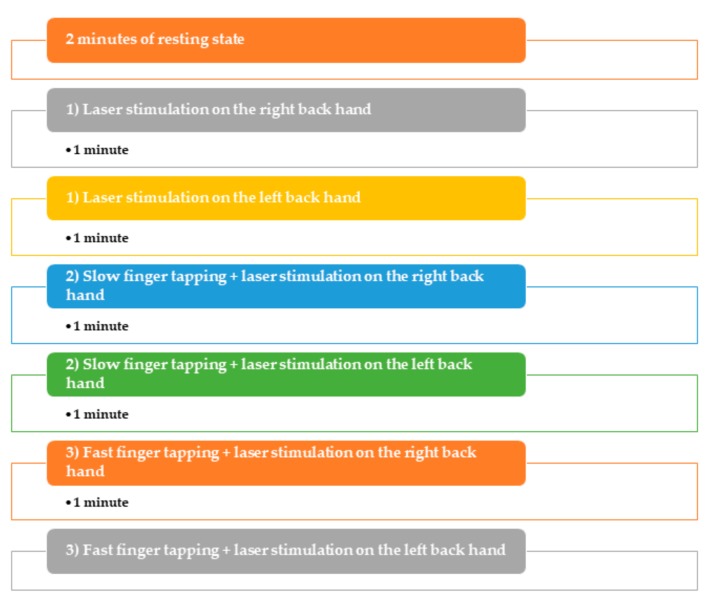
The randomized order of experimental conditions.

**Figure 2 brainsci-10-00190-f002:**
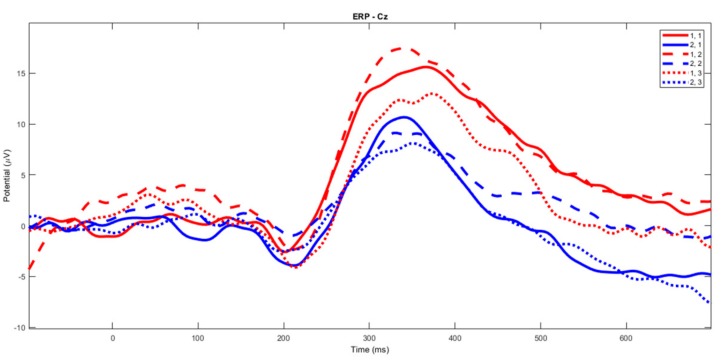
The grand average of the laser–evoked potentials on the right hand stimulation. The control group’s basal condition (1,1), slow finger tapping (1,2) and fast finger tapping (1,3); the fibromyalgia (FM) group’s basal condition (2,1), slow finger tapping (2,2) and fast finger tapping (2,3) (see also Figure 1).

**Figure 3 brainsci-10-00190-f003:**
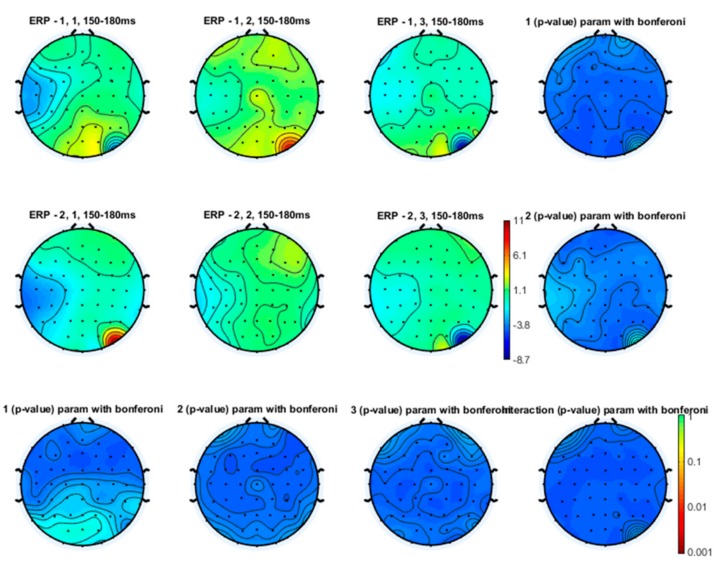
Topographical representation of the N1 wave on the right hand stimulation. The first line of topographical maps represents the N1 amplitude in the baseline condition, slow finger tapping (SFT) and fast finger tapping (FFT) in the control group. The second line of maps represents the N1 amplitude in the baseline condition, slow finger tapping (SFT) and fast finger tapping (FFT) in the patient group. The control group’s basal condition (1,1), slow finger tapping (1,2) and fast finger tapping (1,3); the fibromyalgia (FM) group’s basal condition (2,1), slow finger tapping (2,2) and fast finger tapping (2,3) (see also Figure 1). The results of the repeated measures analysis of variance (ANOVA) corrected with the Bonferroni test are reported.

**Figure 4 brainsci-10-00190-f004:**
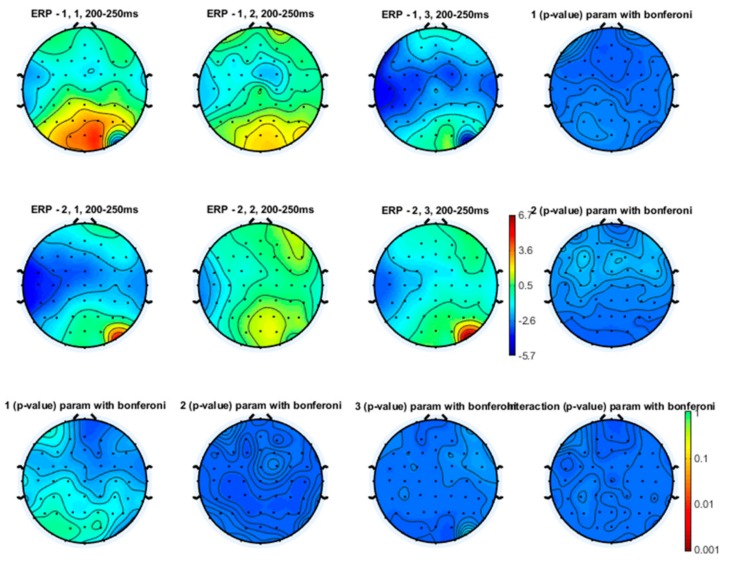
Topographical representation of the N2 component on the right hand stimulation. The first line of topographical maps represents the N2 amplitude in the baseline condition, slow finger tapping (SFT) and fast finger tapping (FFT) in the control group. The second line of maps represents the N2 amplitude in the baseline condition, slow finger tapping (SFT) and fast finger tapping (FFT) in the patient group. The control group’s basal condition (1,1), slow finger tapping (1,2) and fast finger tapping (1,3); the fibromyalgia (FM) group’s basal condition (2,1), slow finger tapping (2,2) and fast finger tapping (2,3) (see also Figure 1). The results of the repeated measures analysis of variance (ANOVA) corrected with the Bonferroni test are reported.

**Figure 5 brainsci-10-00190-f005:**
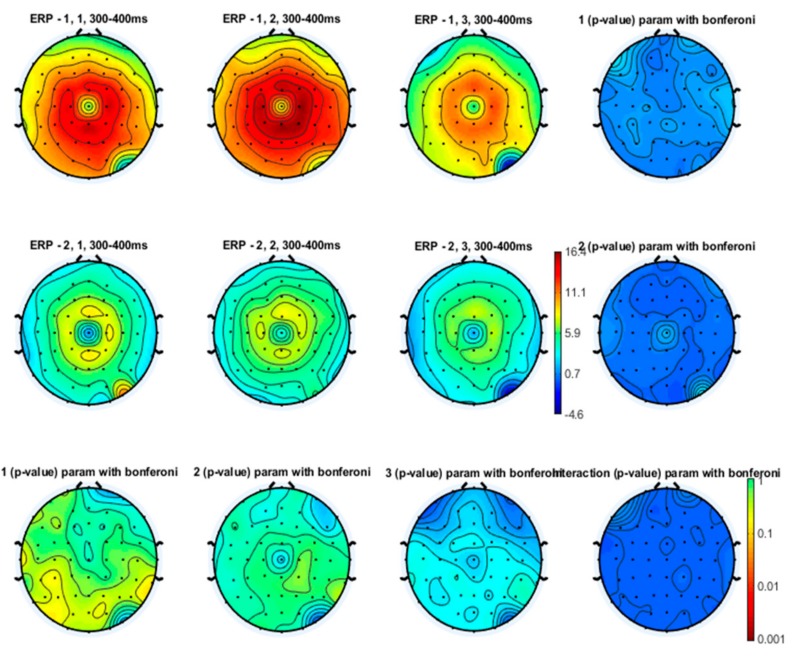
Topographical representation of the P2 component on the right hand stimulation. The first line of topographical maps represents the P2 amplitude in the baseline condition, slow finger tapping (SFT) and fast finger tapping (FFT) in the control group. The second line of maps represents the P2 amplitude in the baseline condition, slow finger tapping (SFT) and fast finger tapping (FFT) in the patient group. The control group’s basal condition (1,1), slow finger tapping (1,2) and fast finger tapping (1,3); the fibromyalgia (FM) group’s basal condition (2,1), slow finger tapping (2,2) and fast finger tapping (2,3) (see also Figure 1). The results of the repeated measures analysis of variance (ANOVA) corrected with the Bonferroni test are reported.

**Figure 6 brainsci-10-00190-f006:**
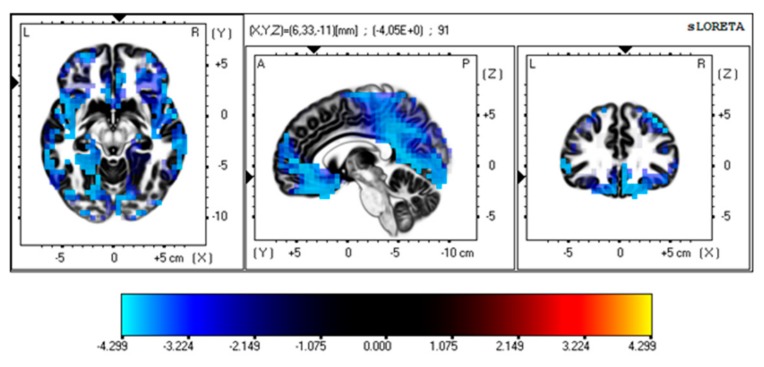
The sLORETA voxels expressing statistical analysis results in the control group between LEPs in the basal condition and during fast finger tapping. The maps express the maximal difference in light blue, corresponding to frontal and limbic regions. For details of the analysis, see Appendix A.

**Figure 7 brainsci-10-00190-f007:**
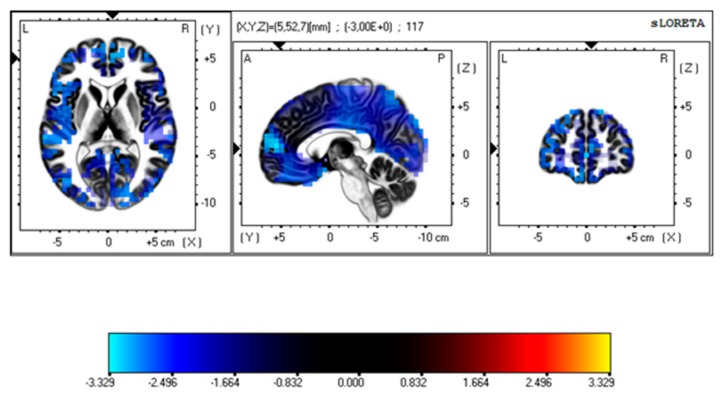
The sLORETA voxels expressing statistical analysis results of the comparison of changes of laser evoked potential (LEP) sources in the basal condition and during fast finger tapping between the controls and the fibromyalgia (FM) patients. The maps express the maximal difference in light blue, corresponding to frontal and limbic regions. For details of the analysis, see Appendix A.

**Figure 8 brainsci-10-00190-f008:**
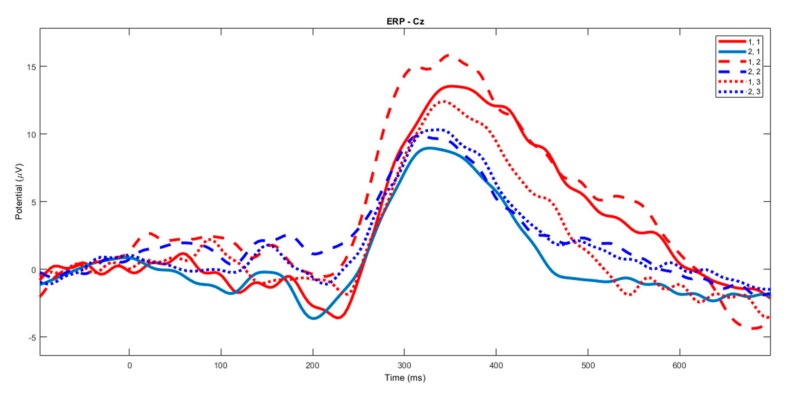
The grand average of the laser-evoked potentials on the left hand stimulation. The control group’s basal condition (1,1), slow finger tapping (1,2) and fast finger tapping (1,3); the fibromyalgia (FM) group’s basal condition (2,1), slow finger tapping (2,2) and fast finger tapping (2,3) (see also Figure 1).

**Figure 9 brainsci-10-00190-f009:**
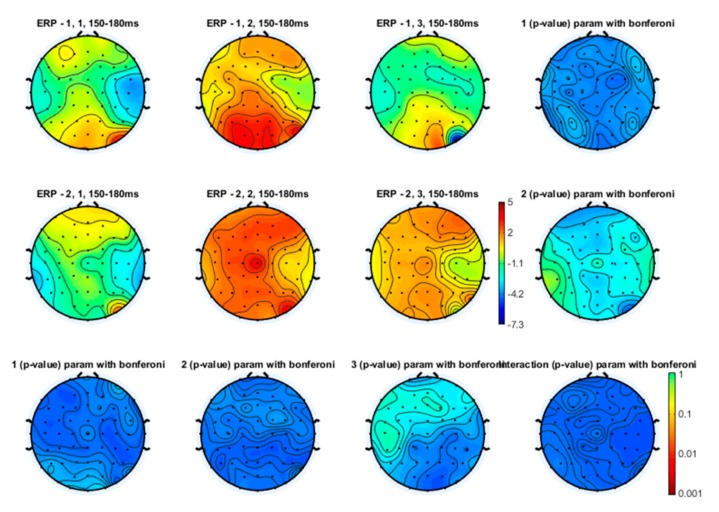
Topographical representation of the N1 component on the left hand stimulation. The first line of topographical maps represents the N1 amplitude in the baseline condition, slow finger tapping (SFT) and fast finger tapping (FFT) in the control group. The second line of maps represents the N1 amplitude in the baseline condition, slow finger tapping (SFT) and fast finger tapping (FFT) in the patient group. The control group’s basal condition (1,1), slow finger tapping (1,2) and fast finger tapping (1,3); the fibromyalgia (FM) group’s basal condition (2,1), slow finger tapping (2,2) and fast finger tapping (2,3) (see also Figure 1). The results of the repeated measures analysis of variance (ANOVA) corrected with the Bonferroni test are reported.

**Figure 10 brainsci-10-00190-f010:**
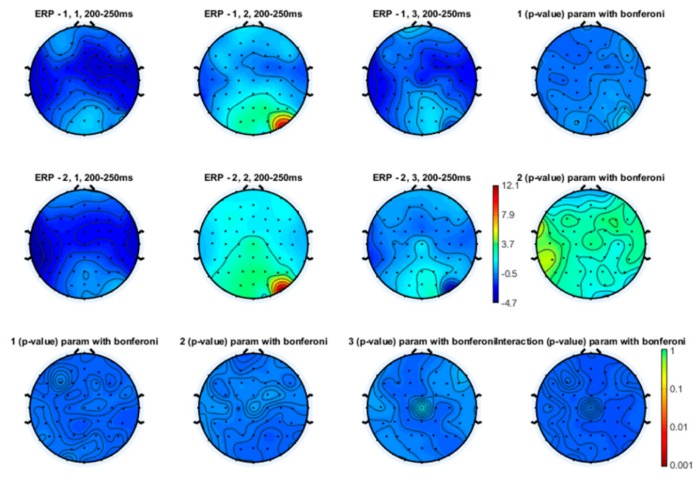
Topographical representation of the N2 component on the left hand stimulation. The first line of topographical maps represents the N2 amplitude in the baseline condition, slow finger tapping (SFT) and fast finger tapping (FFT) in the control group. The second line of maps represents the N2 amplitude in the baseline condition, slow finger tapping (SFT) and fast finger tapping (FFT) in the patient group. The control group’s basal condition (1,1), slow finger tapping (1,2) and fast finger tapping (1,3); the fibromyalgia (FM) group’s basal condition (2,1), slow finger tapping (2,2) and fast finger tapping (2,3) (see also Figure 1). The results of the repeated measures analysis of variance (ANOVA) corrected with the Bonferroni test are reported.

**Figure 11 brainsci-10-00190-f011:**
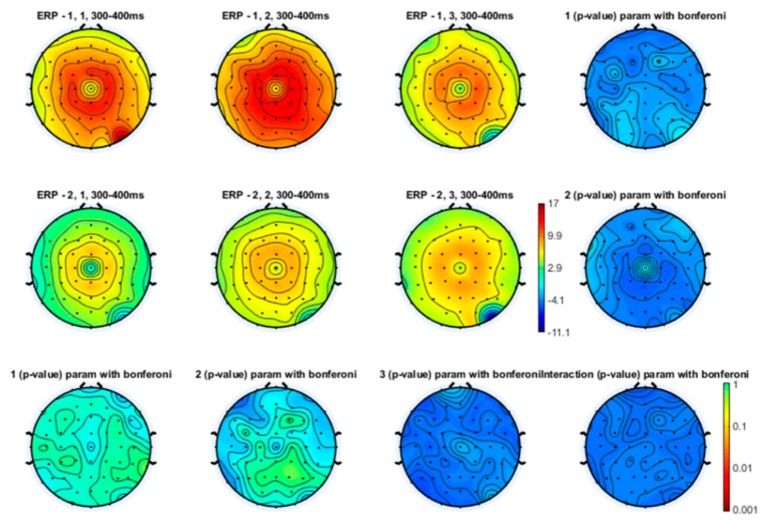
Topographical representation of the P2 component on the left hand stimulation. The first line of topographical maps represents the P2 amplitude in the baseline condition, slow finger tapping (SFT) and fast finger tapping (FFT) in the control group. The second line of maps represents the P2 amplitude in the baseline condition, slow finger tapping (SFT) and fast finger tapping (FFT) in the patient group. The control group’s basal condition (1,1), slow finger tapping (1,2) and fast finger tapping (1,3); the fibromyalgia (FM) group’s basal condition (2,1), slow finger tapping (2,2) and fast finger tapping (2,3) (see also Figure 1). The results of the repeated measures analysis of variance (ANOVA) corrected with the Bonferroni test are reported.

**Table 1 brainsci-10-00190-t001:** Two-way Analysis of Variance (ANOVA) results.

Test of Effects between Subjects					
Dependent Variable: VAS					
Source	Sum of Squares III	Df	Mean Squares	F	Sig.
Correct Model	302079.314	17	17769.371	1.343	0.164
Intercept	4221019.314	1	4221019.314	319.025	0.00
Group	244266.783	2	122133.391	9.231	0.000
Condition	2826.598	5	565.320	0.043	0.999
Group* Condition	15380.405	10	1538.040	0.116	1.000
Error	4419156.555	334	13231.008		
Total	73779578.000	352			
Corrected total	4721235.870	351			

The two-way ANOVA was used to compare the dependent variables between the two groups. Visual analogue scale (VAS): dependent variable; Group and Conditions: factors. Df: degree of freedom, F: F-test, Sig: *p*-value.

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
