# Peer review of "A Simple Pattern of Movement Is Not Able to Inhibit Experimental Pain in FM Patients and Controls: An sLORETA Study"

_brainsci, 2020, doi:10.3390/brainsci10030190_

Round 1
Reviewer 1 Report
This study evaluates the effects of motor cortex activation on pain in fibromyalgia patients vs controls. The changes were analyzed of laser evoked potentials during a finger tapping task (slow and fast). Each finger tapping task was also performed during noxious laser stimulation of both right and left hand.
The results show that a simple and repetitive movement is not able to induce a consistent inhibition of experimental pain evoked by the moving and not moving hand in each group.
The topic is of interest, the experimental design is sound and the results are convincing.
I have the following comments:
-The examined fibromyalgia patients should be better described. Which was the severity level of their pain symptoms? What the degree of their comorbidities? Were they all homogeneous? Were there any differences in the examined parameters in this study in subgroups of patients depending on their pain characteristics in basal conditions?
-In the female patients in their fertile phase of life, was the phase of the menstrual cycle taken into account for the planning of the experiments, i.e., were tests performed in the same relative phase of the cycle in all?
-The English language contains several imperfections. The authors should have their ms revised by a mother tongue person.
The ref list could be implemented in regard to the recent literature on fibromyalgia. In particular the recent classification of primary chronic pain conditions by the IASP group for the ICD-11 should be quoted:
The IASP Classification of Chronic Pain for ICD-11: Chronic Primary Pain, Pain, 160(1)28-37.
Author Response
Dear reviewer,
We are pleased for your comments of this article. I attach the corrections about your suggestion.
Reviewer 1:
- The examined fibromyalgia patients should be better described. Which was the severity level of their pain symptoms? What the degree of their comorbidities? Were they all homogeneous? Were there any differences in the examined parameters in this study in subgroups of patients depending on their pain characteristics in basal conditions?
We thank the reviewer for his useful comments. We completely agree with your suggestion and included this information in the materials and methods. The changes were outlined in yellow color.
- In the female patients in their fertile phase of life, was the phase of the menstrual cycle taken into account for the planning of the experiments, i.e., were tests performed the same relative phase of the cycle in all?
The experimental sessions were all carried out in the same phase of the menstrual cycle for patients with fibromyalgia.
- The English language contains several imperfections. The authors should have their ms revised by a mother tongue person.
We had the text reviewed by a native speaker, as suggested. Thank you.
- The ref list could be implemented in regard to the recent literature on fibromyalgia. In particular the recent classification of primary chronic pain conditions by the IASP group for the ICD-11 should be quoted.
Thanks for suggesting this article, we have included it in our work.
Reviewer 2 Report
The article is very interesting, innovative and well designed.
I just have some observations:
- I think it is important to report how many laser stimulations and recordings were collected and analyzed in both patient and control groups for each session. The article shows the grand average of laser evoked potentials but not the number of single observations.
- Line 34-35 "laser evoked potentials are a useful tool in the study of pain pathways and how they are modulated under different conditions, including current movement". The meaning of this sentence is not clear, maybe the word "how" does not fit.
- Line 148 "FFT+Laser stimulation on the left hand" is written 2 times (in line 147 was already written "FFT+Laser stimulation on the right or left back hand)
Author Response
Dear reviewer,
We are pleased for your comments of this article. I attach the corrections about your suggestion.
The changes were outlined in green color.
- I think it is important to report how many laser stimulations and recordings were collected and analysed in both patient and control groups for each session. The article shows the grand average of laser evoked potentials but not the number of single observations.
Thanks for this clarification, we have entered the indication of the number of laser stimulations delivered for each individual experimental session.
- Line 34-35 “laser evoked potentials are useful tool in the study of pain pathways and how they are modulated under different conditions, including current movement”. The meaning of this sentence is not clear, maybe the word “how” does not fit.
We have corrected the sentence.
- Line 148 “FFT+ laser stimulation on the left hand” is written 2 times (in line 147 was already written “FFT+ laser stimulation on the right or left back hand).
Thank you for pointing out this inaccuracy.